# Methotrexate Alters the Expression of microRNA in Fibroblast-like Synovial Cells in Rheumatoid Arthritis

**DOI:** 10.3390/ijms222111561

**Published:** 2021-10-26

**Authors:** Naoki Iwamoto, Kaori Furukawa, Yushiro Endo, Toshimasa Shimizu, Remi Sumiyoshi, Masataka Umeda, Tomohiro Koga, Shin-ya Kawashiri, Takashi Igawa, Kunihiro Ichinose, Mami Tamai, Tomoki Origuchi, Atsushi Kawakami

**Affiliations:** 1Department of Immunology and Rheumatology, Division of Advanced Preventive Medical Sciences, Graduate School of Biomedical Sciences, Nagasaki University, Nagasaki 852-8102, Japan; furukaka@nagasaki-u.ac.jp (K.F.); yushiro19861105@hotmail.co.jp (Y.E.); toshimasashimizu2000@yahoo.co.jp (T.S.); remis@nagasaki-u.ac.jp (R.S.); masatakau0807@gmail.com (M.U.); tkoga@nagasaki-u.ac.jp (T.K.); shin-ya@nagasaki-u.ac.jp (S.-y.K.); t-igawa1973road@jeans.ocn.ne.jp (T.I.); kichinos@nagasaki-u.ac.jp (K.I.); tamaim@nagasaki-u.ac.jp (M.T.); atsushik@nagasaki-u.ac.jp (A.K.); 2Center for Bioinformatics and Molecular Medicine, Graduate School of Biomedical Sciences, Nagasaki University, Nagasaki 852-8102, Japan; 3Division of Advanced Preventive Medical Sciences, Department of Community Medicine, Graduate School of Biomedical Sciences, Nagasaki University, Nagasaki 852-8102, Japan; 4Department of Physical Therapy, Graduate School of Biomedical Sciences, Nagasaki University, Nagasaki 852-8102, Japan; origuchi@nagasaki-u.ac.jp

**Keywords:** MTX, miR-877-3p, rheumatoid arthritis, fibroblast-like synovial cells, microRNA

## Abstract

We aimed to investigate the effect of methotrexate (MTX) on microRNA modulation in rheumatoid arthritis fibroblast-like synovial cells (RA-FLS). RA-FLS were treated with MTX for 48 h. We then performed miRNA array analysis to investigate differentially expressed miRNAs. Transfection with miR-877-3p precursor and inhibitor were used to investigate the functional role of miR-877-3p in RA-FLS. Gene ontology analysis was used to investigate the cellular processes involving miR-877-3p. The production of cytokines/chemokines was screened by multiplex cytokine/chemokine bead assay and confirmed by ELISA and quantitative real-time PCR. The migratory and proliferative activities of RA-FLS were analyzed by wound healing assay and MKI-67 expression. MTX treatment altered the expression of 13 miRNAs (seven were upregulated and six were downregulated). Among them, quantitative real-time PCR confirmed that miR-877-3p was upregulated in response to MTX (1.79 ± 0.46-fold, *p* < 0.05). The possible target genes of miR-877-3p in RA-FLS revealed by the microarray analysis were correlated with biological processes. The overexpression of miR-877-3p decreased the production of GM-CSF and CCL3, and the overexpression of miR-877-3p inhibited migratory and proliferative activity. MTX altered the miR-877-3p expression on RA-FLS, and this alteration of miR-877-3p attenuated the abundant production of cytokines/chemokines and proliferative property of RA-FLS.

## 1. Introduction

Rheumatoid arthritis (RA) is a systemic autoimmune disease with severe joint inflammation, characterized by progressive bone and cartilage destruction. Not only immune cells such as T cells, B cells, and macrophages but also fibroblast-like synovial cells (FLS) play critical roles in the pathogenesis of RA. FLS have an antiapoptotic property and an invasive phenotype that cause hyperplastic synovial tissue, leading to the destruction of cartilage and bone. The presence of FLS also contributes to the local production of proinflammatory cytokines, chemokines, and proteolytic enzymes that degrade the extracellular matrix [1].

Methotrexate (MTX) is a folic acid antagonist that is widely used as an anchor drug in the treatment of RA [2]. Several pharmacological mechanisms by which MTX attenuates the disease condition of RA have been reported. MTX showed anti-inflammatory effects by inducing extracellular adenosine, which binds to adenosine receptors [3]. MTX also suppressed the proliferation of lymphocytes and neutrophil chemotaxis and inhibited the proliferation of vascular endothelial cells [4,5,6]. Moreover, as a mechanism that is unique to RA pathogenesis, both the induction of apoptosis of RA-FLS and the inhibition of cytokine production from RA-FLS by MTX have been reported [7,8,9]. However, the precise mechanisms of how MTX inhibits cytokine production and exerts antiproliferative effects on RA-FLS remain unknown.

MicroRNAs (miRNAs), small endogenous RNAs that regulate the expression of genes post-transcriptionally, have emerged as key regulators of biologic processes. Relationships between miRNAs and disorders such as malignancies and autoimmune diseases have been reported, and it has been speculated that miRNAs modulate the cellular processes of RA-FLS [10,11,12]. Furthermore, the alteration of miRNA expression by drugs has recently been reported. Sorafenib, an antitumor drug for advanced hepatocellular carcinoma, increased the cellular expression of miR-125a in cultured hepatocellular carcinoma cells, and the upregulation of this miRNA inhibited cell proliferation by suppressing sitruin-7 [13]. In the breast cancer cell line MCF-7, it was reported that the upregulation of miR-33b by lovastatin led to a reduction of cell proliferation, and the global expression profile of miRNAs was altered by 5-fluorouracil; 23 miRNAs were upregulated and 19 were downregulated [14,15]. The alteration of miRNAs by a disease-modifying antirheumatic drugs (DMARD) was also reported: sulfasalazine upregulated miR-136-5p in microglia [16]. Serum miR-339-5p and let-7i-5p were reduced by MTX treatment in RA [17] and relations between several circulating miRNAs and treatment response in RA have been reported [18,19,20,21].

In this study, we examine the alteration of the expression of miRNAs on RA-FLS by MTX. We also seek to elucidate the function of MTX-modulated miRNAs to investigate the potential of miRNA as novel treatment option for RA.

## 2. Results

### 2.1. miR-877-3p is an MTX-Inducible miRNA in RA-FLS

We first performed miRNA microarray scanning to search for MTX-altered miRNAs. The effect of MTX on RA-FLS was confirmed by the suppression of IL-6 production in an ELISA (94.47 vs. 63.76 pg/mL, *p* < 0.05). The microarray analysis identified 13 differentially expressed microRNAs in MTX-treated RA-FLS. Of these, seven miRNAs were upregulated and six were downregulated among all three sets of MTX-treated RA-FLS compared with each control (Figure 1A,B). To validate these microarray findings, we performed a quantitative RT-PCR with additional cultured fibroblasts, and we observed that among the 13 miRNAs, miR-877-3p was consistently upregulated in MTX-treated RA-FLS to 1.784  ±  1.209 (*p* < 0.05) (Figure 1C).

### 2.2. miR-877-3p Transfection Microarrays

The function of a specific miRNA is to suppress the expression of its target mRNA, and thus a gene that is suppressed by the overexpression of that miRNA and promoted by the knockdown of the same miRNA is considered the target of the miRNA. To elucidate the functional consequences of the abnormal expression of miR-877-3p in RA-FLS, we investigated the genes modulated by miR-877-3p and their possible cellular effects in RA-FLS after gene modulation by performing a microarray analysis and subsequent GO analysis.

The results revealed that miRNA transfection with pre-miR877-3p increased the levels of miR-877-3p by 2.69 × 10^4^ -fold compared to the scrambled controls, whereas knockdown with anti-miR-877-3p reduced the expression of miR-877-3p to 0.0093-fold. This indicated successful transfection. We examined the gene expression profiles by performing a microarray analysis: 74 genes were observed to be downregulated by the overexpression of miR-877-3p and upregulated by the knockdown of miR-877-3p (data are available from the corresponding author, N.I., upon reasonable request). These genes were analyzed in the context of GO annotations for the identification of significantly affected GO functional categories.

As shown in Figure 2, the most significant change was observed in categories related to cellular responses to chemical stimuli. Categories such as cell signaling and regulation of the immune system were also highly involved. These results suggested that an alteration of miR-877-3p may be involved in biological processes in RA-FLS.

### 2.3. miR-877-3p Reduced the Productions of GM-CSF and CCL3

We next evaluated the influence of the productions of the cytokines/chemokines that are strongly associated with the pathogenesis of RA. The screening experiments using the multiplex bead assay revealed a reduced production of several cytokines including GM-CSF and tumor necrosis factor-alpha (TNF-α) and chemokines such as CCL3 and CCL7 from pre-miR-877-3p-transfected RA-FLS compared to those from control RA-FLS (Appendix A).

Among the humoral factors that were changed by the overexpression of miR-877-3p as detected by the multiplex bead assay, GM-CSF and CCL3 showed consistent and significant results. The overexpression of miR-877-3p reduced the productions of GM-CSF (4.22 vs. 1.95 ng/mL, *p* < 0.05) and CCL3 (507.9 vs. 316.5 pg/mL, *p* < 0.05) (Figure 3A,B). Moreover, overexpression of mi-877-3p significantly decreased GM-CSF and CCL3 mRNA expression (Figure 3C,D). These results suggested that miR-877-3p inhibit GM-CSF and CCL3 production at the transcription level in RA-FLS.

### 2.4. miR-877-3p Inhibited the Migratory Ability of RA-FLS

To assess whether the deregulation of miR-877-3p affects the migration and proliferation of RA-FLS, we examined the migratory effect by a scratch wound healing assay and the proliferative effect by the marker of proliferation Ki-67 (MKI67) mRNA expression. The overexpression of miR-877-3p resulted in a reduction of cell migration of 30.8% compared to the control (Figure 4A,B). This finding was confirmed by the observed expression of MKI67 mRNA, the marker of cell proliferation. Consistent with wound healing assay, MKI67 expression was decreased by the overexpression of miR-877-3p (0.59 ± 0.23-fold, *p* < 0.05, Figure 4C).

## 3. Discussion

This is the first study to demonstrate that MTX alters the expression of microRNAs in RA-FLS. Our findings demonstrated that (1) the expression of miR-877-3p was increased by the MTX treatment of RA-FLS, and (2) the upregulation of miR-877-3p led to decreased productions of cytokines and chemokines and decreased cell migration.

Cytokines and chemokines play a crucial role in the pathogenesis of RA, and one of the key players that secretes abundant amounts of these humoral factors is FLS. In the present study, miR-877-3p downregulated the expression of GM-CSF and CCL3 in RA-FLS at both the protein and mRNA level. GM-CSF, which was originally defined as a hemopoietic growth factor due to its ability to form colonies of granulocytes and macrophages, was reported to be increased in the synovial fluid and blood from patients with RA, and the administration of GM-CSF to RA patients led to disease flares [22,23,24]. Based on those findings, GM-CSF inhibition has been considered as a potential therapeutic option for RA [25]. CCL3, a member of the CC subfamily, is also known as macrophage inflammation protein 1-a (MIP-1a). CCL3 induces a variety of proinflammatory activities such as leukocyte chemotaxis [26]. Zhang et al. reported that the expression of CCL3 was increased in the peripheral blood and synovial fluid of RA patients [27]. The inhibition of the production of IL-6 from RA-FLS by treatment with MTX is well known, but the effects of MTX on the production of other cytokines and chemokines have not been established [8]. Our present findings suggest that the MTX treatment of RA-FLS might also inhibit the production of these cytokines and chemokines via an upregulation of miR-877-3p. However, the influence of an overexpression of miR-877-3p on signaling pathways such as the Janus kinase/signal transducer and activator of transcription (JAK/STAT) pathway and the phosphoinositide 3-kinase/v-akt murine thymoma (PI3K/AKT) signaling pathway that are deeply related to cytokines/chemokines production, was not investigated in the present study. We should analyze the modification of the signaling pathway by overexpression of miR-877-3p in the future. As other possible regulation by miR-877-3p, miR-877-3p contains almost only UC nucleotides, indicating a pyrimidine-rich sequence, and a functional analysis of the cellular pathway by performing in silico studies showed that a pyrimidine-rich sequence might be involved in lysine degradation, RNA transport, and spliceosome [28]. Thus, these functions might be changed on RA-FLS by overexpression of miR-877-3p caused by the MTX treatment.

Another hallmark of RA pathogenesis is the increased migration activity and resistance to apoptosis of the FLS cells, which result in pannus formation and the abundant production of proinflammatory cytokines. The modulation of this “tumor-like” behavior of RA-FLS is thought to improve the inflammation in RA-affected joints. In the present study, the migration and proliferation of RA-FLS was decreased by a forced expression of miR-877-3p. Several reports showed that because MTX prevents the de novo synthesis of pyrimidine and purine (which are required for DNA and RNA synthesis), MTX inhibits the cellular proliferation of lymphocytes depending on the alteration of the intracellular reactive oxygen species (ROS) levels [29,30,31,32]. Considering the effect of miR-877-3p on cell migrative and proliferative activity demonstrated in the present study, there is a possibility that MTX inhibits the proliferation of RA-FLS by both an upregulation of miR-877-3p and the prevention of the synthesis of DNA and RNA.

One miRNA regulates several genes transcriptionally and post-transcriptionally, indeed, in the present study 74 genes were suspected to be regulated by miR-877-3p in RA-FLS. A GO analysis revealed that these genes were related to cell signaling and the regulation of the immune system. Synovial cytokine network including cell signaling among immune cells such as macrophages, T cells, and FLS are considered as one of the main pathogenesis of RA [33]. Furthermore, the dysregulation of immune systems like a regulatory T cell dysfunction is also related to developing RA [34]. As shown by the result of our GO analysis, miR-877-3p has the potential to regulate RA-FLS though these regulations.

Limitation of this study is that there was no non-RA disease samples. To investigate whether this effect of MTX is specific to RA or not, an analysis of the effect of MTX using fibroblast from non-RA such as osteoarthritis is needed. Moreover, MTX is also used for cancer treatment and there is a possibility that an antiproliferative effect caused by MTX-microRNA modulation is present in cancer cells. Another limitation is that other miRNAs besides miR-877-3p were not investigated. Other candidate miRNAs revealed by miRNA microarray might be regulated by MTX and modify the cytokine and chemokine production and proliferation in RA-FLS. Similarly, multiplex bead assay showed an inhibition of several cytokines and chemokines by overexpression of miR-877-3p (Appendix A), however, not all these altered cytokines and chemokines were investigated in detail. 

This is the first study to establish a mechanism link between the anti-inflammatory, antimigratory effect of MTX on RA-FLS and the upregulation of miR-877-3p. Another important finding of our study is that the upregulation of miR-877-3p has the potential to provide a foundation for a novel treatment of RA via the modulation of RA-FLS. Although the functions of miR-877-3p have not been widely addressed, several studies of miR-877-3p have been reported. Zhu et al. reported that enforced expression of miR-877-3p increased the expression of P16, which inhibits the proliferation of bladder cancer and Chen et al. reported that overexpression of miR-877-3p abolished the migration and invasion of cervical cancer cell lines induced by HOXD-AS1 [35,36]. Another study showed the promotion of osteoblast differentiation of MC3T3-E1 cells by overexpression of miR-877-3p. [37]. Like these examples, the functions of miRNAs differ by cell types, especially cancer cells versus noncancer cells [38,39]. The role of miRNAs might be different, even in the MTX treatment in RA, between FLS and serum. In fact, the reduction of serum miR-339-5p and let7i-5p after MTX treatment have been reported by Cunningham et al. [17].

In conclusion, our results revealed that MTX altered the microRNA expression profiles in RA-FLS. We speculate that miR-877-3p might be a downstream effector of MTX in the suppression of cytokine/chemokine production and the invasive phenotype. This knowledge may be useful for the development of novel therapeutic strategies based on other treatments able to boost the cellular reservoir of miR-877-3p.

## 4. Materials and Methods

### 4.1. Isolation of FLS and Treatment by MTX

Each patient provided a signed consent form to participate in the study, which was approved by the Institutional Review Board of Nagasaki University (IRB approval no. 112072361). All of the RA patients fulfilled the 2010 American College of Rheumatology (ACR)/European League against Rheumatism (EULAR) classification criteria for RA or the 1987 ACR classification criteria for RA at the time of orthopedic surgery [40,41]. We obtained synovial tissues from patients with RA at the time of orthopedic surgery, and then FLS from the RA patients were isolated from synovial tissues as described (*n* = 7) [42].

The RA-FLS were plated at a cell density of 1 × 10^5^ in six-well plates. After 70% confluence, cells were starved in a Dulbecco’s modified Eagle’s medium (DMEM) containing 0.5% fetal bovine serum (FBS) (all from Gibco, Basel, Switzerland) for 24 h, and then the cells were treated with MTX (Sigma, St. Louis, MO, USA) (1 μM) for 48 h.

### 4.2. RNA isolation and Quantitative Real-Time PCR Analysis

A mirVana miRNA Isolation kit (Ambion/Applied Biosystems, Foster City, CA, USA) was used for the isolation of total RNA. Specific single TaqMan miRNA assays (Ambion/Applied Biosystems) were used to measure the expression levels of selected miRNAs in a model light cycler 1.5 (Roche Diagnostics, Indianapolis, IN, USA). The expression of the U6B small nuclear RNA (RNU6B) was used as an endogenous control for the normalization of the data. In the analysis of the expression of specific mRNA, quantification of each mRNA was performed by SYBR Green real-time PCR (Qiagen, Hilden, Germany), as previously described [43]. The primers were obtained from Takara Bio (Tokyo, Japan), and the primer sequences are shown in Table 1. Expression of GAPDH was used as endogenous control to normalize the data. For relative quantification, the comparative threshold cycle method was used [44].

### 4.3. MicroRNA and DNA Microarray Assay Analyses

The profiling of the miRNA expression of RA-FLS treated with MTX was established by applying the SurePrint G3 Human miRNA, 8 × 60 K (release 18.0) microarrays containing 1887 human miRNA oligonucleotide probes (Agilent Technologies, Santa Clara, CA, USA). The DNA microarray analysis was performed using the whole human genome DNA microarray SurePrint G3 Human Gene Expression, 8 × 60 K (v. 2.0) microarrays (Agilent Technologies). All procedures were carried out according to the manufacturer’s recommendations. The microarray data were analyzed by GeneSpring software ver. 12.5.0 or 12.6.1 (Agilent Technologies, Santa Clara, CA, USA). The raw signals were log2 transformed and normalized using the percentile shift normalization method: the values were set at the 90th percentile for the miRNA microarray and at the 75th percentile for the DNA microarray.

A gene ontology (GO) analysis was performed using the list of genes that were upregulated by more than twofold by the downregulation of miR-877-3p and those that were downregulated by less than twofold by the overexpression of miR-877-3p; these genes were suspected target genes of miR-877-3p on the RA-FLS. We analyzed the target genes using the GO annotation in the GO database (http://geneontology.org accessed on 1 September 2021) for the evaluation of the genes’ functions.

### 4.4. Transfection Experiments

RA-FLSs were transfected with a synthetic precursor miRNA of miR-877-3p (pre-miR) (PM13557), with inhibitors of miR-877-3p (anti-miR) (AM13557) at the concentration of 50 uM, or with scrambled controls (pre-miR/anti-miR negative control #1; all from Ambion/Applied Biosystems) with the use of Lipofectamine 2000 reagent (Invitrogen, Carlsbad, CA, USA). The transfection efficiency of pre/anti-mir-877-3p was confirmed by a TaqMan-based real-time polymerase chain reaction (RT-PCR).

### 4.5. Multiplex Cytokines/Chemokines Bead Assay and ELISA

RA-FLS transfected with pre-miR-877-3p or scrambled controls were stimulated with interleukin-1beta (IL-1β) (10 ng/mL) (R&D Systems, Minneapolis, MN, USA), and the supernatants were collected after 48 h. To obtain the profile of the production of cytokines and chemokines, we then performed a multiplex cytokine bead assay using MILLIPLEX MAP human cytokine/chemokine magnetic bead panel 1-premixed 38 Plex (Millipore, Billerica, MA, USA) kits according to the manufacturer’s instructions. Proteins were detected by an enzyme-linked immunosorbent assay (ELISA) using ELISA kits specific for IL-6, granulocyte macrophage colony-stimulating factor (GM-CSF), and CC chemokine ligand (CCL) 3 according to the manufacturer’s instructions (all from R&D Systems).

### 4.6. Wound Healing Assay

To analyze the wound healing capacity of the RA-FLS, we used an in vitro model of wound healing (CytoSelect™ 24-well wound healing assay, Cell Biolabs, San Diego, CA, USA). The kit consists of 24-well plates containing 12 proprietary treated plastic inserts. The inserts create a wound field with a defined gap of 0.9 mm for measuring the migration and proliferation rate of cells.

RA-FLS (5 × 10^4^) were seeded and transfected with pre-miR-877-3p or scrambled controls. At 24 h after cultivation in DMEM containing 10% FBS under stimulation with lipopolysaccharide (LPS) (10 ng/mL) (Sigma), the cells were stained with a cell stain solution and the migration distance was photographed. Wound healing images were analyzed using ImageJ (National Institutes of Health, Bethesda, MA, USA). The rate of wound closure was calculated as follows: percent closure (%) = the surface area of the migrated cells into the wound area (migrated cell surface area) / the surface area of wound area (total surface area) × 100.

### 4.7. Statistical Analyses

The statistical significance of the differences between groups was evaluated by Student’s paired *t*-tests. All data are expressed as the mean  ±  standard deviation (SD). *p*-values < 0.05 were considered significant. GraphPad Prism software (GraphPad, San Diego, CA, USA) was used for the statistical analyses.

## Figures and Tables

**Figure 1 ijms-22-11561-f001:**
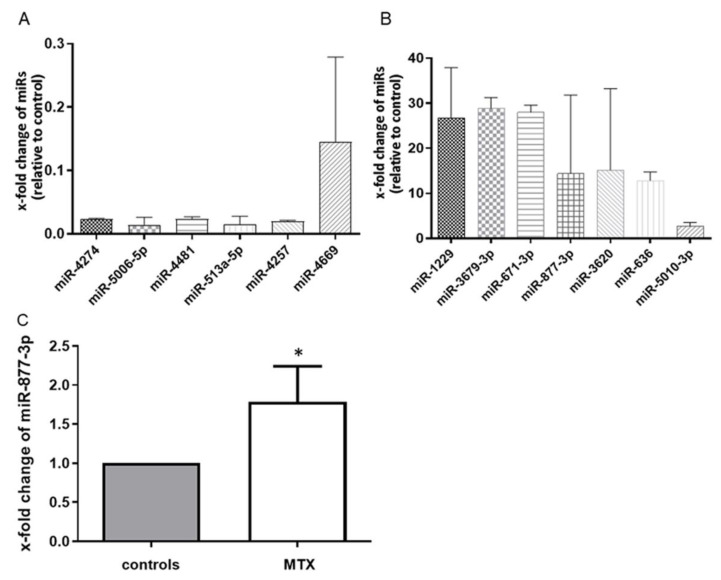
MTX altered the global expression profile of RA-FLS (*n* = 3). (**A**) the six microRNAs in the downregulated group. (**B**) the seven microRNAs in the upregulated group. Values are the means of three pairs (RA-FLS treated with MTX vs. untreated RA-FLS). We defined “altered miRNA” that were upregulated or downregulated in all three pairs. (**C**) the expression of microRNA-877-3p (miR-877-3p) in RA-FLS treated with MTX as determined by TaqMan-based RT-PCR. The expression of miR-877-3p in RA-FLS treated with MTX was determined relative to untreated RA-FLS, which was set as 1. MiR-877-3p was upregulated in RA-FLS after 48 h MTX treatment (*n* = 7). Values are presented as means ± SD. * *p* < 0.05 versus control.

**Figure 2 ijms-22-11561-f002:**
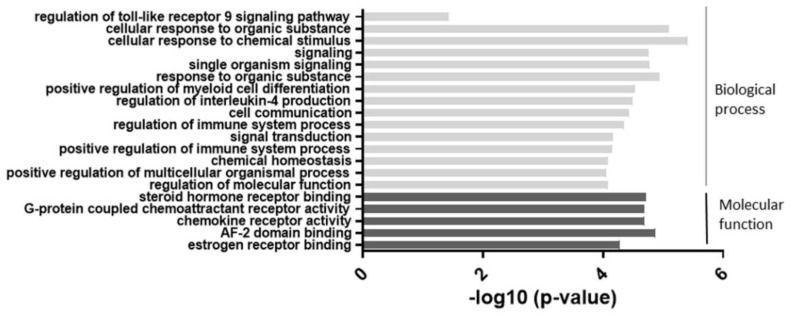
The results of the gene ontology (GO) enrichment analysis of the 74 differentially expressed genes which were suspected miR-877-3p targeted genes in RA-FLS. The enrichment scores (−log 10) of significant enrichment in GO terms are shown. The vertical axis represents the GO categories. The horizontal axis represents the enrichment scores. The *p*-value indicates the significance of the GO term correlated to the conditions. The smaller the *p*-value, the more significant the GO term is. MiR-877-3p affected mainly biological processes such as signaling and cellular responses in RA-FLS.

**Figure 3 ijms-22-11561-f003:**
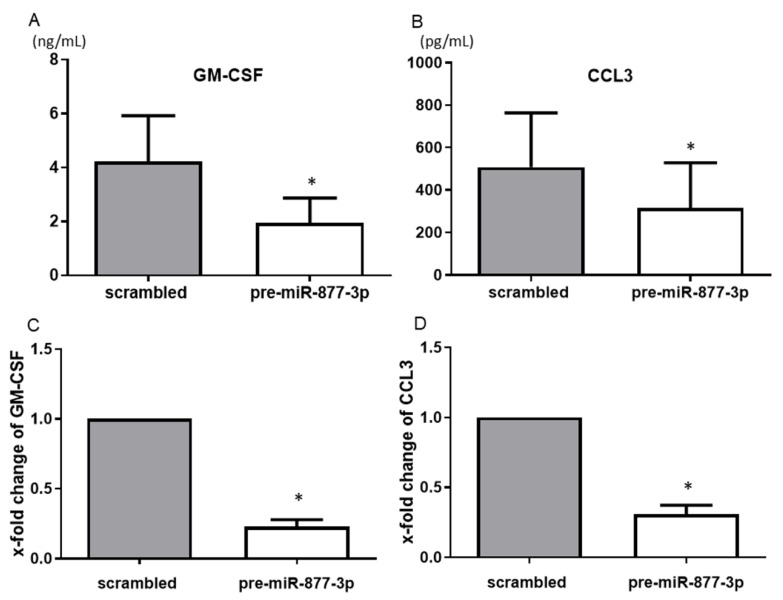
The suppression of the production of chemokines from RA-FLS by the overexpression of miR-877-3p. The transfection of RA-FLS (*n* = 3–4) with precursor miR-877-3p (pre-miR-877-3p) for 48 h decreased the productions of GM-CSF (**A**) and CCL3 (**B**) in the culture supernatant compared to the scrambled-RNA-transfected controls, as determined by ELISA. At the mRNA level, transfection of RA-FLS (*n* = 4) with pre-miR-877-3p for 48 h decreased the levels of GM-CSF (**C**) and CCL3 (**D**) compared to scrambled-RNA-transfected controls, as determined by SYBR Green real-time PCR. Values are means  ±  SD. * *p* < 0.05 vs. scrambled controls.

**Figure 4 ijms-22-11561-f004:**
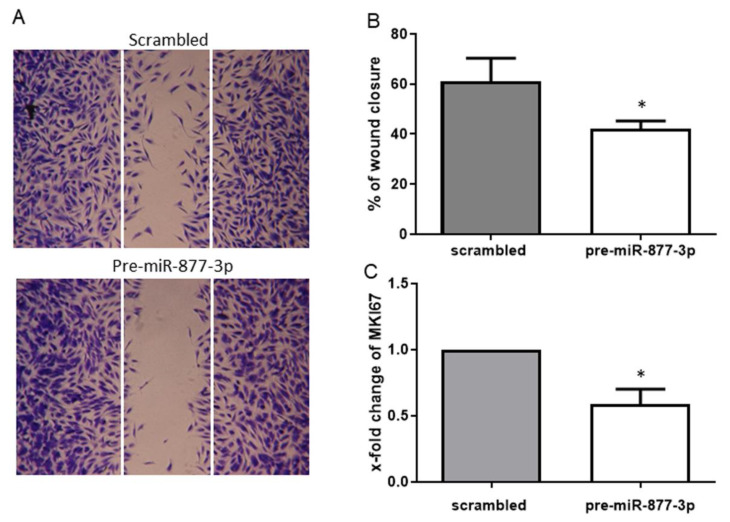
The overexpression of miR-877-3p inhibited the migration and proliferation of RA-FLS. (**A**) representative light microscopy images of RA-FLS. FLS monolayers 24 h after wounding are shown. (**B**) cell migration was analyzed for 24 h (*n* = 4). (**C**) expression of MKI67 in pre-miR-877-3p-transfected RA-FLS (*n* = 4). Values are means  ±  SD. * *p* < 0.05 vs. scrambled controls.

**Table 1 ijms-22-11561-t001:** SYBR Green primers used for real-time PCR.

GMCSF forward	5′-CATGATGGCCAGCCACTACAA-3′
GMCSF reverse	5′-ACTGGCTCCCAGCAGTCAAAG-3′
CCL3 forward	5′-CGGCAGATTCCACAGAATTTCATA-3′
CCL3 reverse	5′-AGATGACACCGGGCTTGGAG-3′
MKI67 forward	5′-CCATATGCCTGTGGAGTGGAA-3′
MKI67 reverse	5′-CCACCCTTAGCGTGCTCTTGA-3′
GAPDH forward	5′-GCACCGTCAAGGCTGAGAAC-3′
GAPDH reverse	5′-TGGTGAAGACGCCAGTGGA-3′

## Data Availability

Not applicable.

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
