# Peer review of "Methotrexate Alters the Expression of microRNA in Fibroblast-like Synovial Cells in Rheumatoid Arthritis"

_ijms, 2021, doi:10.3390/ijms222111561_

Round 1

Reviewer 1 Report

  1. The manuscript by Iwamoto et al is aimed at studying MTX altered the miR-887-3p expression on RA-FLS, and suggest that alteration of miR-887-3p attenuated the abundant production of chemokines/cytokines and proliferative property of RA-FLS. However, some points should be clarified.

    1. Please confirmed “The RA-FLS were plated at a cell density of 1 × 104 in six well plates. (2.1 Isolation of FLS and treatment by MTX)” The number of cells is unnormal, it seems too less to perform experiment.
    2. In fig 1C. What happen with the other miRNA? There were effects on them? Could the authors show that results? In general, these results are lost all over the figures.
    3. Figure 2 should be discussed more detail in discussion part.
    4. What was the rationale of selecting RA-FLS for MTX treatment rather than other disease or cancer types since MTX has anti-proliferation and apoptosis effect?
    5. The effect of MTX in non-RA disease needs to be determined and compare with RA-FLS.
    6. Where are the primer sequences in table 1 that mentioned in supplemental data?
    7. How does author normalize the q-PCR data, the relative quantification and the equation used in 2.2?
    8. For 2.4, author should provide the miRNA sequence for the transfection and the efficiency of the miRNA mimic and inhibitor.
    9. How many cells were seeded for wound healing for the study? Does the media contain FBS or not? Please be specify the detail of experiment.
    10. At result 4.1 figure 1, the error bar isn’t too large for analyze? How about other target miRNA? Were there any effects on them for C? Author should show these results to support of selecting miR-877-3p? Since mrR-3620 has similar result as miRNA-877-3p, why not choosing this target?
    11. In figure 1, where is the internal control for comparing the fold change for A and B? Without control group to determine the downregulated and upregulated group in the figure is not properly demonstrated.
    12. Author should provide the comparison for RA and non-RA disease when treating miR-877-3p to clarify the reason.
    13. In result 4.2 transfection microarrays, how does the fold number calculate for miR-877-39 by 2.69x104 fold to the control? How about other miRNA showing in figure 1?
    14. In figure 4A, how does author quantify the wound closure? By what kind of method or software?
    15. In discussion, “downregulated the expression of GM-CSF and CCL3 in RA-FLS at both protein and mRNA level”, where is the protein level result? Author should provide the data to support for the discussion part.
    16. Since author only showed MTX inhibits cytokines and chemokines by increase miR-877-3p expression, how about other miRNA expression? Does these miRNAs have no effects on cytokines and chemokines?
    17. In discussion, “MTX inhibits the cellular proliferation of lymphocytes depending on the alternation of the intracellular ROS level”. Is there any solid results or experiments conducted to support this statement? Author should provide the actual result of ROS activity and cell proliferation assay to support the statement.
    18. Discussion needs to be elaborated by discussing of all endpoints analyzed, particularly discussing the limitation of the study.
    19. Since author obtained the FLS samples from patients, how does these isolations of FLS cells share the similar or same commend for the study? Is every patient share the same background? How many samples were obtained from patients for the study?
    20. Please, specify the company of SYBR green and TaqMan used including reference.
    21. Could author describe more details of figure 2 and how to obtain this figure?
    22. What is the logical relationship between of apoptosis, proliferation, migration and inflammation inhibited by MTX through miR-877-3p? Author should design some experiments to clarify, otherwise the study will just be a list of some data.

Reviewer 2 Report

In this brief report Iwamoto et al. evaluated the effect of methotrexate on the expression of microRNAs in fibroblast-like synovial (FLS) cells isolated from RA patients. The authors found that methotrexate upregulated the expression of 7 miRNAs and downregulated 6 miRNAs. They also reported that overexpressed miR-877-3p inhibited migratory and proliferative activity of RA-FLS and decreased the production of GM-CSF and CCL3 (MIP-1a). Although the paper is well structured and written, there are some issues that have to be addressed.

  1. It is not clear from how many RA patients were isolated FLS cells and whether these patients received some treatment before the surgery.
  2. The isolation procedure is not cited correctly. In the cited reference (20) this procedure is not described.
  3. What was the reason to select miR-877-3p for further analyses? From Figure 1 could be seen that there are much better candidates.
  4. Data in the supplementary Table S1 show that CCL4 and CCL7 (related to the inflammation) are also altered significantly. Why these data are not discussed?
  5. There are other studies reported for influence of methotrexate on the miRNAs in RA patients (DOI: 10.1016/j.phrs.2021.105887; DOI: 10.3899/jrheum.170266; DOI: 10.2174/1389201019666180417155140; DOI: 10.1007/s10067-018-4380-z; DOI: 10.18632/aging.203201), which are neglected. They are neither included in the Introduction nor discussed.

Round 2

Reviewer 1 Report

No comments

This manuscript is a resubmission of an earlier submission. The following is a list of the peer review reports and author responses from that submission.

Round 1

Reviewer 1 Report

Dear Authors,

This is the interesting and first report about miR-887-3p that was detected and altered the expression level in MTX treated RA-FLS. Additionally, the report showed the over-expression of miR-887-3p decreased the production of GM-CSF and CCL3, and the over-expression of miR-887-3p inhibited migratory activity of RA-FLS.

There are several questions below;

  1. Page 4, line 143: Can you show qRT-PCR data about 7 up-regulated and 6 down-regulated microRNAs detected in microRNA-array? If you can show, please add supplementary data.
  2. Page 4, line 173: Did you confirm the alteration of mRNA or protein level in signaling pathway related molecule such as JAK-STAT, PI3K etc? 

Sincerely,

Reviewer 2 Report

The manuscript by Iwamoto et al is aimed at studying MTX altered the miR-887-3p expression on RA-FLS, and suggest that alteration of miR-887-3p attenuated the abundant production of chemokines/cytokines and proliferative property of RA-FLS. However, some points should be clarified.

Major point:

  1. The association between for MTX and RA has been reported in previous reports. For example: Serum miRNA Signature in Rheumatoid Arthritis and “At-Risk Individuals”. This part should be mentioned and discussed.
  2. The miR-887-3p regulation is important in this study. The author should discuss the possible signaling pathways on miR-887-3p
  3. Are any consist of sequences of miR-887-3p? This part should be mentioned and discussed.
  4. The dramatic decrease in GM-CSF and CCL3 production in the transfected cells is difficult to imagine. The authors would need to provide the transfection efficiency, which is using primary synovial fibroblasts.
  5. The authors should examine the cell proliferation of miRNA inhibitor in RA.
  6. To assess whether the down-regulation of miR-887-3p inhibited the migratory ability of RA-FLS should use more experiment to illustrate. The author should explore the miR-887-3p  could also be involved in possible gene expression.

Minor point:

  1. How about the treating concentration of miRNA inhibitor?
  2. The method of transfection miRNA inhibitor should be described.
